

# Automatic contraction of unstructured tensor networks

Adam S. Jermyn⋆

Center for Computational Astrophysics, Flatiron Institute, New York, NY 10010, USA

⋆ adamjermyn@gmail.com

## Abstract

The evaluation of partition functions is a central problem in statistical physics. For lattice systems and other discrete models the partition function may be expressed as the contraction of a tensor network. Unfortunately computing such contractions is difficult, and many methods to make this tractable require periodic or otherwise structured networks. Here I present a new algorithm for contracting unstructured tensor networks. This method makes no assumptions about the structure of the network and performs well in both structured and unstructured cases so long as the correlation structure is local.


# 1   Introduction

A central problem of classical statistical physics is that of calculating the partition function, which encodes thermodynamic quantities and statistical correlations as a weighted sum over all possible configurations of a system [1]. Because of the Euclidean path integral this is also a central problem in quantum statistical physics [2]. This problem is also closely related to the Bayesian inference problem, and methods which solve the one are readily applied to the other [3].

In infinite systems the partition function is not always computable or even well-defined [4], but when the state space is finite there are no such problems. Being computable does not, however, mean that it is straightforward or tractable to compute. For example, lattice models have local structure. In these systems the state space is an outer product of many local spaces, and so the number of terms in the partition function grows exponentially in system size [5]. This makes a naive numerical evaluation of the partition function impractical.

A variety of general stochastic methods have been developed to tackle this problem, from Nested Sampling [6] to the Metropolis-Hastings algorithm [7] and Wang-Landau sampling [8], as well as numerous variants on each of these. These methods may be applied to arbitrary finite models, but they make no guarantees of convergence or performance. Indeed a well-known problem of such methods is that they may silently fail, entirely and without warning missing the most relevant regions of parameter space [9].

By contrast there are also algorithms with much more limited scope but much more certain performance. The most famous of these is the transfer matrix [10], which takes advantage of the fact that in one-dimensional models with short-range interactions the partition function factors into a product of matrices. For models with such structure this algorithm produces results to high precision with run time that scales at worst linearly in the size of the system and at worst cubically in the size of the local state space.

A recent generalization of the transfer matrix is the tensor network. Tensor networks are multigraphs wherein each node is a tensor and each edge is a contraction between indices on the tensors it connects [11]. This structure allows tensor networks to encode correlations in more complex systems, and notably allows them to represent arbitrary discrete lattice models. It is therefore crucial to develop the means to efficiently manipulate such networks as this would make a tremendous array of problems numerically tractable, from disordered lattices to simulating quantum computers [12] to complex biological and chemical models [13–15].

In certain special cases, most notably trees (acyclic graphs), a tensor network may be efficiently summed as a series of matrix multiplications. Thus the transfer matrix method is just a special case of tensor tree summation. In most cases, however, directly summing a tensor network is intractable because each time a pair of indices is contracted the resulting tensor has greater rank than either of the input tensors. The computational cost of working with a tensor scales exponentially in its rank, and so direct summation typically comes with exponential cost in system size.

Methods have been developed to address this challenge. In small systems there is often room for optimizing the order of contraction [16], which serves to reduce the effective base of the exponential. There has also been some work on approximating the local environment of a

tensor in a network, akin to a numerical mean-field theory, and that has produced promising results in manipulating small networks [17] In larger and even infinite systems with regular crystalline structure a variety of hand-crafted methods have arisen [18–24]. These perform incredibly well, with polynomial run time in system size for in finite systems and accurate results even near critical points in infinite systems. Unfortunately they are often specific to a given model, and are not easy to extend. They also all require periodic or otherwise regular lattice structures and in some cases impose additional symmetries [25], making use of this either to save on storage requirements or to impose constraints on the correlation structure.

In this work I present an algorithm for contracting large finite tensor networks which places no a priori constraints on their structure. This means that it can be applied as a general tool to study correlations in local finite lattice systems with discrete state spaces, a niche currently only occupied by stochastic methods.

I begin in Section 2 with a discussion of the problem of rank explosion. In Section 3 I describe how this may be mitigated by using tensor tree decompositions, which efficiently represent high-rank tensors to within a controlled error threshold. In order to contract pairs of tensor trees it is necessary to eliminate cycles and to do so efficiently. This is the core of the algorithm and is discussed in detail in Section 4. Section 5 then puts the pieces together and describes the overall method of contracting tensor networks using these tools. In Section 6 I then demonstrate the performance of this algorithm in many real-world examples. While it does not come with any guarantees of run time efficiency, in practice it exhibits polynomial scaling in system size far from critical points and exponential scaling near them. It also converges to the desired accuracy, being a controlled approximation method. Finally I have released a software implementation of this algorithm along with several related methods, and the details of this implementation are given in Appendix A.

## 2   Rank Explosion

Tensor networks span many different communities within mathematics and physics and so it is best to be clear about nomenclature. The rank of a tensor is the number of indices it possesses. The dimension of a given index is the range over which it spans (i.e. the number of elements it introduces into sums), or equivalently the dimension of the dual space whose members map the tensor to a tensor possessing all but the specified index. The shape of a tensor is the collection of its index dimensions. Finally the size of a tensor is the number of elements it contains, which is equal to the product of the dimensions of its indices.

The phenomenon of rank explosion occurs when successive contraction operations on average increase the ranks of tensors in a network. When this is not accompanied by a commensurate decrease in the typical index dimension it results in an explosion in the number of elements the network contains. Even if all indices have dimension two, which is the lowest non-trivial dimension, a contraction which results in a tensor of rank greater than both of the ranks of the input tensors always results in a network of at least as many elements. If the rank increases by more than one, or the typical dimension is greater than two or the two input tensors were not of the same size, then the number of elements generically increases. This is a problem because in numerical algorithms the bottleneck is usually manipulating and storing these elements, and so a dramatic increase in their number is typically accompanied by a dramatic decrease in performance.

As an example consider the tensor network depicted in Figure 1. The network is drawn with Penrose notation, with shapes representing tensors and lines representing indices [26]. Where lines attached to different tensors connect those indices are to be contracted. In this network there are two tensors of rank 4 with one contraction specified. After performing this

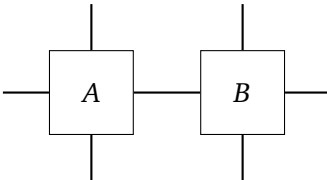

Figure 1: Two tensors are shown in Penrose notation. This network specifies a single contraction over a single pair of indices between the two tensors, as well as several external indices on each.

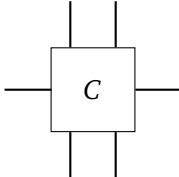

Figure 2: The same network is shown as in Figure 1 after contracting $A$ and $B$ along their shared edge.

contraction the network appears as in Figure 2, with one tensor of rank 6. More generally, whenever two tensors of ranks $r_1 \geq r_2$ sharing $k$ links are contracted, the resulting rank is $r_1 + r_2 - 2k$. This exceeds both input ranks when $2k < r_2$. In a $d$-dimensional square lattice model, for instance, $k = 1$ and $r = 2d$ (see Figure 3), so for $d > 1$ this is a problem. The case of $d = 1$ reduces to transfer matrices.

For some networks this is only an apparent problem because subsequent contractions result in a net decrease in rank, or because the rank increases may be halted by careful choice of the contraction sequence [16]. The network shown in Figure 4 has this property. This network is just a 2D lattice model with just two tensors in the vertical direction. Contracting along vertical lines results in an increase in rank, but once all such contractions have been done the model is one-dimensional and may be contracted with no further increases in rank. This is actually generically true for $d > 1$ lattice models, but the rank at which the process halts is proportional to the cross-sectional size of the system along all but the largest dimension, and so may be prohibitively large. This is closely related to the problem faced by DMRG methods



Figure 3: A 2D lattice model partition function is shown as a tensor network on a $5 \times 5$ square lattice. Each tensor has rank 4 and there are $k = 1$ links between any pair of adjacent tensors.

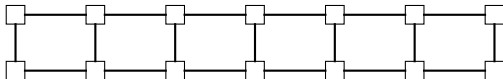

Figure 4: A 2D lattice model partition function is shown as a tensor network on a $7 \times 2$ square lattice. This model does not suffer from rank explosion because it is effectively one-dimensional.

Figure 5: Shown is a rank-6 tensor (left) along with its factorization into a tensor tree (right).

in $d > 1$, where it is possible to incorporate higher dimensions at the cost of run time which is exponential in their extent [27].

## 3 Tensor Trees

One way to avoid the problem of rank explosion is to devise efficient methods for representing high-rank tensors. This problem has received considerable attention from a variety of angles [28–31] and a general theme of hierarchical decomposition has emerged. Such decompositions are advantageous over tensor-train decompositions in cases where the correlations are not local in one dimension [32], and are preferable to sparse tensor schemes because the tensors arising in statistical physics are rarely sparse.

The most well-studied such a decomposition is the tensor tree [30], in which one factors a high-rank tensor into a tree (acyclic network) of lower-rank tensors, as shown in Figure 5. Components of the tensor may then be evaluated efficiently as a series of matrix multiplications. While decompositions cannot improve the representation of all tensors[1], those with local structure in the correlations between their indices can be compressed dramatically [28, 33].

A key feature of tensor tree decompositions is that their efficiency depends heavily on the choice of tree. Significant work has gone into making context-specific choices [13, 32]. Recently automated methods of determining optimal or near-optimal choices have been developed [34]. These methods use the correlation structure of the tensor to infer which indices ought to be near one another on the tree, and so in effect are always context-aware.

## 4 Cycle Elimination

Tensor trees may be efficient for storing tensors, but they be contracted efficiently? That is, given two tree tensors is there an efficient way to produce a new tree tensor representing their contraction? There are two cases in which the answer is unambiguously yes. First, when

---

[1]Otherwise all strings could be compressed, in violation of the pidgeonhole principle.

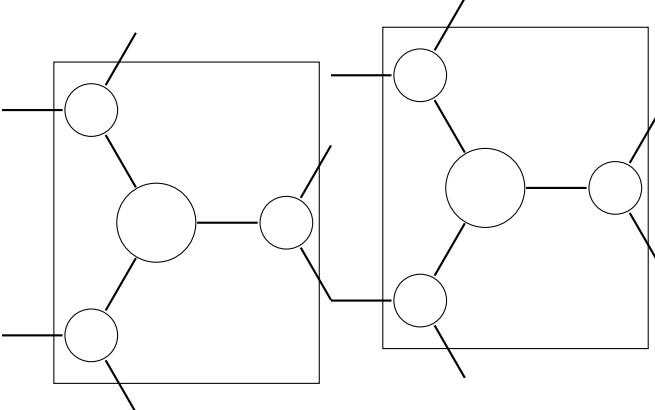

Figure 6: A contraction between two tensor trees along one index pair is shown. For clarity each tree has been placed in a box.

the trees in question are to be contracted along only a single index pair the contraction may be done without any computation at all. For example consider the tensor network shown in Figure 6. There are two tensor trees, shown in boxes, with a single contraction specified. The overall network is already a tree, so the contraction may be performed by simply identifying the network as a single tree tensor. No actual computation is needed.

The second case in which contracting a pair of tree tensors can definitely be contracted efficiently is when the trees are aligned [28]. That is, where the subgraph of each tree containing the indices to be contracted is identical in configuration to the equivalent subgraph on the other tree *with the same index labelling*. In this case there exists a contraction sequence such that at every stage there is at least one cycle consisting of just two tensors. At every stage this sequence reduces the rank of the resulting trees, so it may be carried out without rank explosion. This is illustrated in Figure 7, which from left-to-right shows the process of contracting aligned tensor trees in two different cases. By contrast Figure 8 shows examples of misaligned pairs of tensor trees, for which there is no obviously optimal contraction sequence.

A natural question at this stage is whether or not it is possible to arrange for all contractions to be of one of these two forms. The answer, unfortunately, is no. It is certainly not possible to arrange for them to all be of the first form because that form cannot handle networks with cycles. The second form can handle cycles, but a general network will not always permit repeated contractions of this sort. This is because an aligned contraction leaves no freedom as to the structure of the tree, and so many networks which begin with all trees aligned no longer have this property after just a few contractions. In panel (b) of Figure 7, for instance, a tensor tree which connects to both the two left-most indices and the two-rightmost indices of the final tree is not aligned with it, even though it may well have been aligned with the two trees we began with. A method for handling misaligned trees is therefore necessary.

The fundamental difficulty with misaligned trees is that they possess large cycles. As defined above, an aligned tree pair has a contraction sequence which possesses only cycles of length two, and so may be directly contracted without generating tensors of increasing rank. This is not true for a misaligned tree pair, and naively contracting the cycles which arise in such pairs rapidly increases the ranks of the resulting tensors. To see this consider the the cyclic tensor network shown in Figure 9. This network is symmetric with respect to $i \rightarrow i+1$ modulo 7, and so the first contraction may be performed along any edge. For simplicity we pick the $6-7$ edge, and the result after this contraction is shown in Figure 10. The tensor which resulted from this contraction is now of rank 4, one greater than either of the input ranks. If this tensor is then contracted with one of those on either side the rank increases further to 5, as shown in Figure 11. Each time a tensor is contracted with another in this cycle

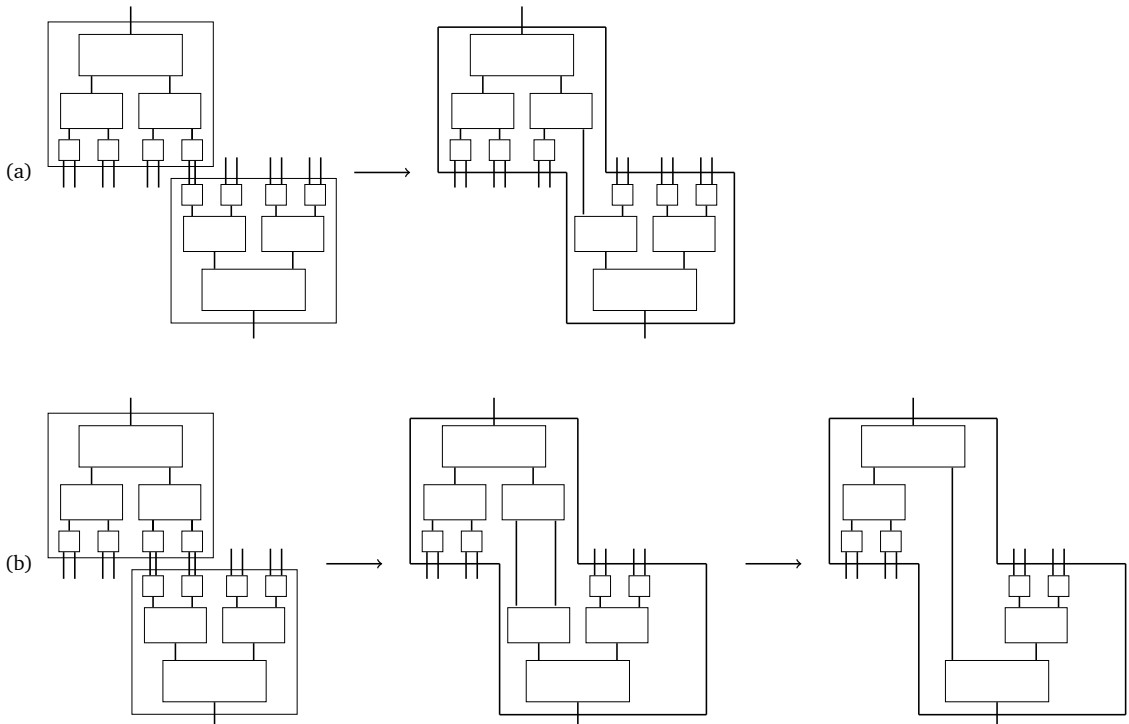

Figure 7: From left to right the contraction of a pair of aligned tensor trees is shown. Cases (a) and (b) correspond respectively to contractions which reach one and two layers deep in each tree. In the end the result is another tensor tree. Note that at each stage there is a pair of indices to be contracted which form a cycle containing just two tensors.

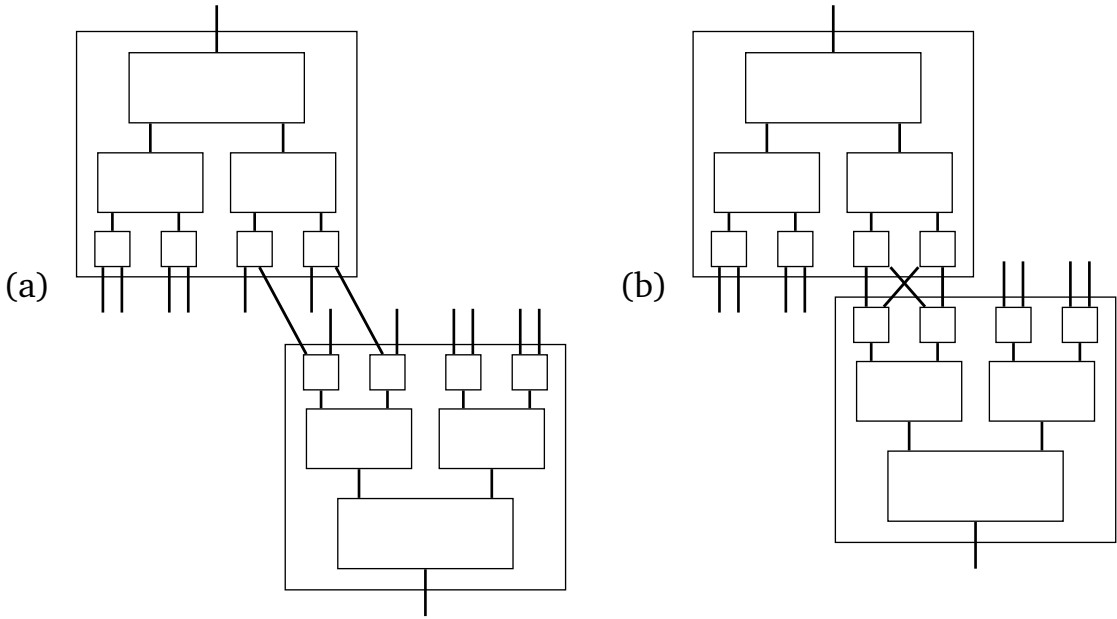

Figure 8: Two examples of misaligned trees are shown. Importantly there is no contraction sequence which avoids creating intermediate tensors of rank larger than that with which the components of the tree began.

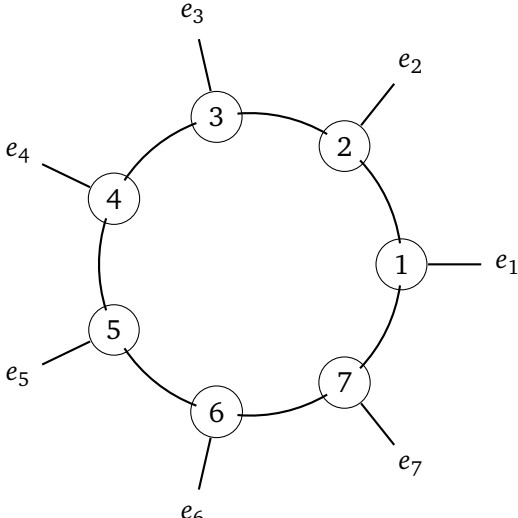

Figure 9: A cyclic tensor network is shown with external indices $\{e_i\}$ and tensors $\{i\}$ for $i \in \{1..7\}$.

the rank increases by at least one. Indeed the situation is worse than that: the final tensor which results must have rank 7 because there are 7 external indices! This is something that no amount of finessing the contraction order can avoid.

One solution is to avoid contracting cycles at all. While this does not sound like much of a solution, recall that the goal is to contract two tensor trees into a new tensor tree. This only needs to be a tree, and so must possess no cycles, but there is no specification as to how that goal is achieved. Instead of contracting cycles then the solution is to *unravel* them. For example consider once more the cyclic network shown in Figure 9. The reason that tensor 7 is a part of this cycle is because two of its indices lead to other tensors in the cycle. The same is true of tensor 6. Figure 12 shows just the portion of this cycle in the immediate vicinity of these two tensors. If they could be rearranged so that both external indices were on one tensor and both indices connecting to the cycle were on the other then one of these tensors would not be in the cycle at all.

This may be achieved by first contracting along the $6-7$ edge as shown in Figure 13. This results in a rank 4 tensor $T$ as before. This tensor may be interpreted as a matrix by defining the composite flattened indices $(i, j)$ and $(k, l)$, where $i$ and $j$ are the indices leading to $e_6$ and $e_7$ and $k$ and $l$ are those leading to tensors 1 and 5. With this,

$$M_{(i,j),(k,l)} = T_{ijkl}. \tag{1}$$

This matrix may then be factored using the singular value decomposition as

$$M_{(i,j),(k,l)} = U_{(i,j),m}\Sigma_{mn}V^{\dagger}_{n,(k,l)}, \tag{2}$$

where $U$ and $V$ are unitary and $\Sigma$ is a diagonal matrix with real non-negative entries [35]. In order for $U$ and $V$ to be unitary the dimensions $d_n$ and $d_m$ of the indices $n$ and $m$ respectively must be

$$d_n = d_i d_j \tag{3}$$
$$d_m = d_k d_l. \tag{4}$$

The singular value decomposition may also be done approximately with an error threshold $\epsilon$ by eliminating singular values below the threshold. This also eliminates the corresponding

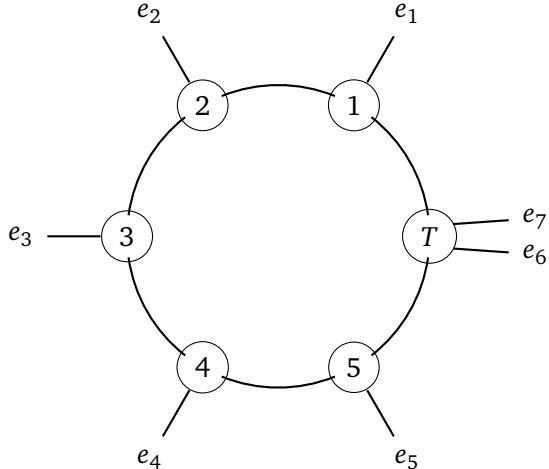

Figure 10: The tensor network in Figure 9 is shown after contracting along the $6-7$ edge. The tensor which results is labeled $T$ and has rank 4, an increase of one from either of tensors 6 or 7.

columns in $U$ and $V$ as well as the associated rows and columns in $\Sigma$ [35]. The result is that $d_n < d_i d_j$ and $d_m < d_k d_l$. Because of this $U$ and $V$ are no longer unitary, as they are no longer square and hence cannot be invertible.

After any rank reduction $\Sigma$ may be absorbed into both $U$ and $V$ as

$$M_{(i,j),(k,l)} = U'_{(i,j),m} V'^{\dagger}_{m,(k,l)}, \tag{5}$$

where

$$U'_{(i,j),m} = U_{(i,j),m} \sqrt{\Sigma}_{mn} \tag{6}$$

and

$$V'_{(k,l),m} = V_{(i,j),m} \sqrt{\Sigma}_{mn}. \tag{7}$$

Note that $\sqrt{\Sigma}$ is perfectly well defined because $\Sigma$ is diagonal. Finally, each of $U'$ and $V'$ may be interpreted as tensors by disassociating the composite indices, as in

$$A_{ijm} = U'_{(i,j),m} \tag{8}$$

and

$$B_{mkl} = V'_{(k,l),m}. \tag{9}$$

This produces the factored result shown in Figure 14, which is shown in the broader context in Figure 15. The cycle has one fewer tensor, with tensor $B$ residing outside of the loop and holding the external indices which were originally held by tensors 6 and 7. In effect these indices have been pinched off. Importantly, there has been no increase in rank except for the intermediate tensor $T$, but in this process every intermediate tensor of that sort is immediately broken down in rank after being formed and so this process carries no risk of rank explosion.

Note that the above procedure of flattening and then contracting two rank-3 tensors, performing an approximate singular value decomposition, and un-flattening the resulting matrices to recover two rank-3 tensors is very similar to the approximation method at the heart of DMRG [36]. The only difference is that here we flatten different sets of indices from the ones we un-flatten, whereas in DMRG the same sets of indices are used in both operations.

This process can clearly be repeated until the cycle becomes a tree. The final step before this occurs is shown in Figure 16. After unravelling with respect to tensors 2 and 3, the network

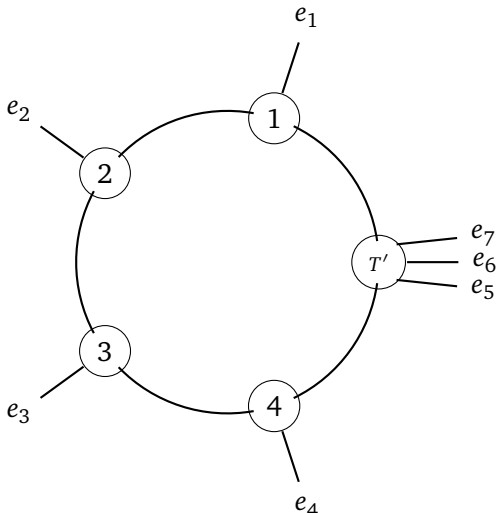

Figure 11: The tensor network in Figure 10 is shown after contracting along the $5-T$ edge. The tensor which results is labeled $T'$ and has rank 5, an increase of one from the larger of the two input tensors.

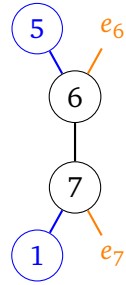

Figure 12: A portion of the cycle from Figure 9 is shown. Tensors 6 and 7 are the focus of this portion, and edges leading away from them towards the rest of the cycle are shown in blue connecting to blue circles (tensors) while those leading out of the cycle are shown in orange.

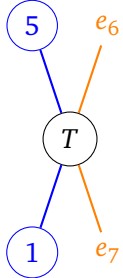

Figure 13: A portion of the cycle from Figure 10 is shown. Tensor $T$ is the focus of this portion, and edges leading away from them towards the rest of the cycle are shown in blue connecting to blue circles (tensors) while those leading out of the cycle are shown in orange.

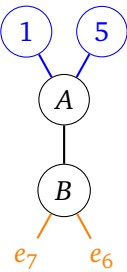

Figure 14: The result of factoring the tensor $T$ in Figure 13 is shown. The new tensors $A$ and $B$ are the focus, and edges leading away from them towards the rest of the cycle are shown in blue connecting to blue circles (tensors) while those leading out of the cycle are shown in orange. Note that $A$ is contained in the cycle because it connects to tensors 1 and 5 while $B$ is not in the cycle, connecting only to $A$ and the external indices $e_6$ and $e_7$.

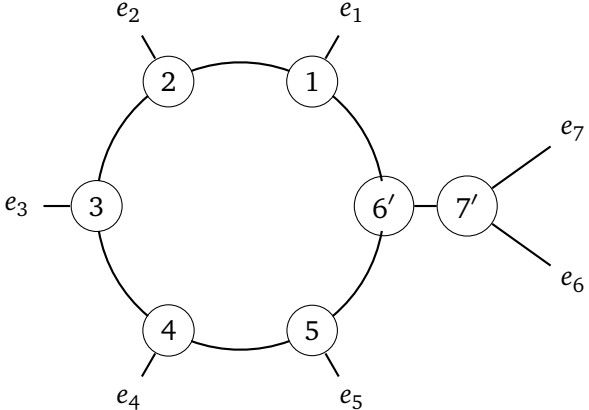

Figure 15: The tensor network in Figure 9 is shown after one unravelling stage, such that $e_6$ and $e_7$ are no longer held by tensors in the cycle. Note that all of the tensors in this network are still rank 3.

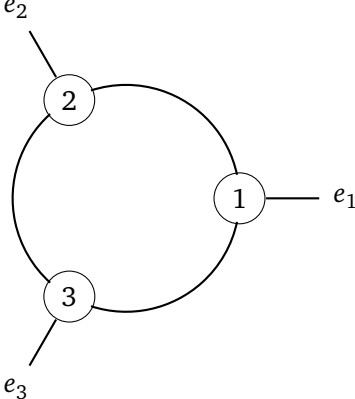

Figure 16: A cyclic tensor network is shown with external indices $\{e_i\}$ and tensors $\{i\}$ for $i \in \{1..3\}$. This is the penultimate result of the cycle unravelling process.

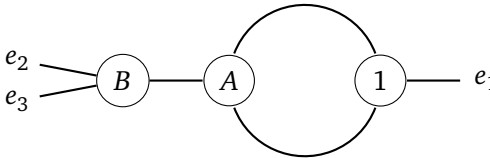

Figure 17: A cyclic tensor network is shown with external indices $\{e_i\}$ and tensors $\{i\}$ for $i \in \{1..3\}$. This is the penultimate result of the cycle unravelling process.

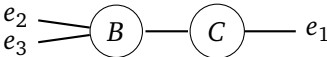

Figure 18: Shown is the final result of unravelling the cycle from Figure 16.

contains three tensors in a row as shown in Figure 17. At this stage all that remains is to contract tensor $A$ with tensor 1 and the result will be a tree, shown in Figure 18.

This method of unravelling cycles is appealing because it is both local and recursive, but it is possible to do better. In particular, it is often the case that a given tree contraction involves several cycles. In this case the order in which indices are removed from a given cycle is crucial because some choices may help to shorten other cycles while others may lengthen them. There is no obviously correct way of accounting for this short of testing a number of alternatives which scales exponentially in network size, and so at this time such effects are best incorporated as a heuristic. This may be done by defining a utility function which accounts for all of the cycles in a network and at every stage select the index-swap (unravel) operation which optimizes this. The details of the one such function which works well in practice are included in Appendix A.4.

## 5 Contracting Networks

With these pieces in place, the process of contracting a tensor network is fairly straightforward. The network is first initialised, and all tensors are internally cast into the form of tensor trees. A contraction sequence is then chosen, typically by heuristic for large networks (see Appendix A.5) because identifying optimal contraction orders takes exponential time in the size of the network [16]. This sequence is then carried out. When a contraction involves no cycles the involved tensor trees are simply concatenated. When the tensor trees involved are aligned the contraction is carried out by iteratively contracting cycles composed of two tensors. Finally, when the tensor trees involved are misaligned the contraction is carried out via cycle elimination as described in the previous section. Once the contraction is complete the contraction sequence may then be updated, and the process proceeds until there are no more contractions to be performed.

At this stage the tensor network is a tensor forest, namely a collection of tensor trees with no connections between them. In the case of a partition function this forest has no external indices, as the partition function is just a scalar. More generally, tensor networks representing $N$-point correlation functions contract to forests with $N$ external indices. Regardless of its origin, the resulting forest permits straightforward evaluation of any of the elements of the tensor network. Upon specifying the element of interest on the external indices, a series of matrix multiplications yields that element.

# 6 Numerical Experiments

This section details various numerical experiments that were performed using the methods introduced in this work. These span a wide array of models, from a dilute Ising spin glass with no regular structure to lattice models, in one and two dimensions, including both regular and disordered models, and including those with periodic as well as open boundary conditions. These were done with the PyTNR library, and both the implementation and the experiments are included in the release of this library, detailed in Appendix A.1. All timing was performed on 10 cores of an Intel Skylake processor, and all times reported are CPU time, counted across all cores used as applicable. In this section the convention that $k_B T = 1$ is used. Furthermore in this section the symbol $Z$ always refers to the partition function:

$$Z = \sum_{\text{states}} e^{-E(\text{state})}. \tag{10}$$

## 6.1 1D Ising Model

To begin consider the 1D Ising model with Hamiltonian

$$H = J \sum_{i=1}^{N} s_i s_{i+1} + h \sum_{i=1}^{N} s_i, \tag{11}$$

where $s_{N+1}$ is identified with $s_1$ so that the boundaries are periodic. The partition function in this case is

$$Z = \sum_{s_1 = \pm 1} \sum_{s_2 = \pm 1} \cdots \sum_{s_N = \pm 1} e^{-J \sum_{i=1}^{N} s_i s_{i+1} - h \sum_{i=1}^{N} s_i}. \tag{12}$$

In PyTNR this is represented by the tensor network shown in Figure 19. The tensors labeled $J$ are each just the matrix

$$M(J) = \begin{bmatrix} e^{-J} & e^{J} \\ e^{J} & e^{-J} \end{bmatrix}. \tag{13}$$

Likewise those labeled $h$ are just the vector

$$v(h) = \begin{bmatrix} e^{-h} \\ e^{h} \end{bmatrix}. \tag{14}$$

Finally, the tensors labeled $s_i$ for $i \in \{1, 2, ..., 5\}$ are just the Kronecker delta tensors $\delta_{jkl}$ which are one when all three indices are equal and zero otherwise. This choice of notation allows us to clearly separate the state space on each side from the interactions between sites.

This model is a useful one to test against because it has several limits with known analytic results. For instance, when $h = 0$ the free energy is

$$F = -\ln Z = -\ln \left[ (2 \sinh(J))^N + (2 \cosh(J))^N \right] \tag{15}$$

[37], where $N$ is the number of sites. Figure 20 shows the free energy PyTNR computes in this limit as a function of the number of sites and for several $J$, along with the residual versus the exact answer and the time required for the computation. These results were produced using the entropy contraction sequence heuristic (see Appendix A.5), no tree optimization (see Appendix A.6), and an SVD truncation accuracy of $10^{-3}$. The error is well below the truncation accuracy, lying near the machine floating point precision. Models with $J > 0$ prefer antiferromagnetic ordering, and so as expected exhibit an oscillatory dependence on the parity of the number of sites.



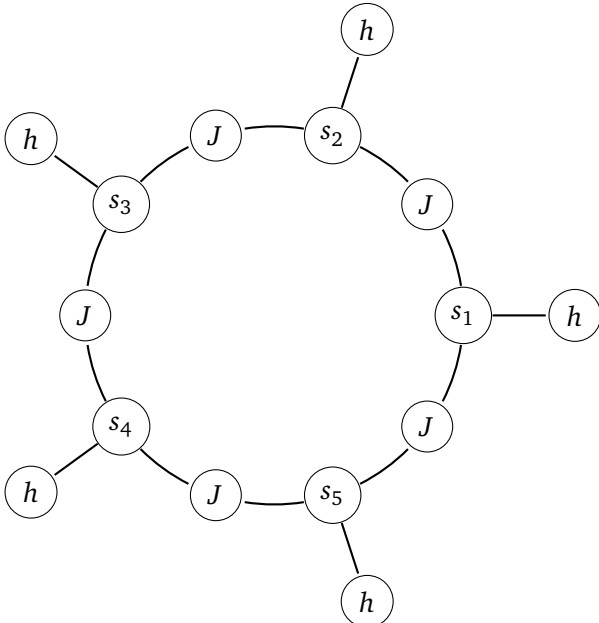

Figure 19: A tensor network representing the 1D Ising model is shown with periodic boundary conditions and $N = 5$ sites. The tensors labeled $J$ encode the interactions, those labeled $h$ encode the onsite term, and those labeled $s_i$ are rank-3 Kronecker delta tensors which enforce the condition that each interaction involving any given spin sees the same spin as each other such interaction.

In general larger models require more run time, but this trend is a polynomial in system size and shows little dependence on $J$, so the computations remain tractable even for large systems. This may be understood because the contraction of a 1D tensor network with $N$ sites reduces to $N$ matrix multiplications of fixed size, so the runtime might be expected to be linear. In fact I find runtime scaling like $N^{1.3}$, which is due to the overhead of our heuristics selecting a contraction ordering.

The opposing limit is less interesting, but provides a useful test nonetheless. In this case $J = 0$, so

$$F = N \ln (2 \cosh(h)) \tag{16}$$

[37]. Figure 21 shows the free energy PyTNR computes in this limit as a function of the number of sites and for several $h$, along with the residual versus the exact answer. These results were produced using the entropy contraction sequence heuristic (see Appendix A.5), no tree optimization, and an SVD truncation accuracy of $10^{-3}$. There is no coupling between sites so there are no finite size effects, and the result is just a flat line in each case. Note that the model was still initialized with the $J$ tensors shown in Figure 19, so this lack of coupling is something that PyTNR computed, rather than being pre-specified. The error is again of order the machine floating point precision. Once more larger models require more run time, but this trend is a polynomial in system size and shows little dependence on $J$, so the computations remain tractable even for large systems.

As a final one-dimensional example, consider the disordered 1D Ising model in which each of $h$ and $J$ are drawn independently per site from identically distributed random normal distributions with unit variance. This model does not have an analytic solution to compare against and is not translation symmetric, and so while the transfer matrix method is applicable many other tensor network methods are not. Figure 22 shows the free energy per site for this model as a function of the number of sites. These results were produced using the entropy contraction

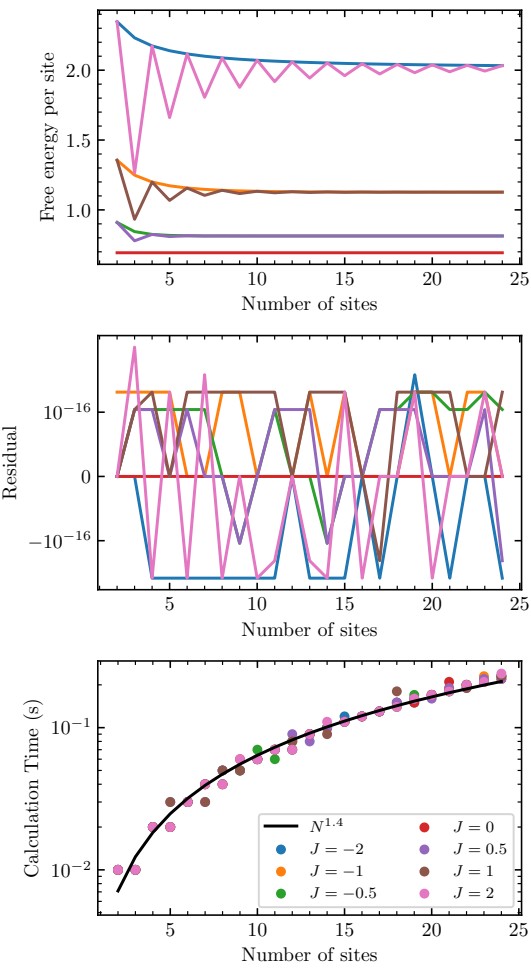

Figure 20: The free energy per site from PyTNR (top), residuals (middle) and run time (bottom) for the 1D Ising model with nearest neighbor interactions is shown as a function of the number of sites for a variety of interaction strengths $J$. Note that models with $J > 0$, which prefer antiferromagnetic ordering, exhibit a dependence on the parity of the number of sites.

sequence heuristic (see Appendix A.5), no tree optimization, and an SVD truncation accuracy of $10^{-3}$.

Each point in Figure 22 represents a distinct sample drawn from the distribution characterizing the model. Note that the variation between neighbouring points decreases with increasing system size. This is expected because larger systems in effect average over a larger number of replicas of the system. Once more the run time is almost independent of the sample, and only shows a dependence on system size.

## 6.2 2D Ising Model

The next example of interest is the 2D Ising model. This is described by the Hamiltonian

$$H = J \sum_{\langle ij \rangle} s_i s_j + h \sum_i s_i, \tag{17}$$

where $\langle ij \rangle$ denotes all nearest-neighbour pairs and $i$ and $j$ index over the entire lattice. The corresponding partition function may be represented by the tensor network shown in Fig-

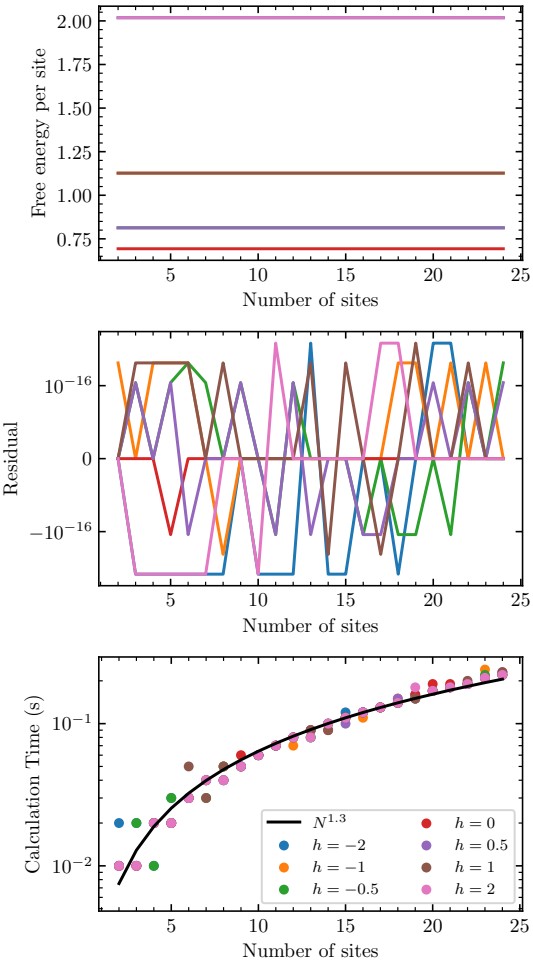

Figure 21: The free energy per site from PyTNR (top), residuals (middle) and run time (bottom) for the 1D Ising model with nearest-neighbor interactions is shown as a function of the number of sites for a variety of on-site energies $h$.

ure 23, where the tensors labeled $h$ and $J$ are as before and the tensors labeled $s$ are rank 5 Kronecker delta tensors.

Except for the trivial case of $h = 0$ this model only permits a closed-form solution in the limit of infinite system size, so our focus will be on how finite size effects decay as the systems become larger. Figure 24 shows the free energy PyTNR computes as a function of the number of sites and for $h = 0$ and several $J$, along with the residual versus the asymptotic result for $N \to \infty$ and the computation time required. These results, and all other results in 2D, were produced using the replica loop contraction sequence heuristic, tree optimization, and an SVD truncation accuracy of $10^{-3}$. The lattice dimensions were chosen to have aspect ratios no greater than 3 and to cover a range of $N$ without large gaps.

As in the 1D case, the free energies of systems with $J \leq 0$ show little dependence on system size, with just a small perturbation that decays as a power law in $N$. The positive $J$, by contrast, show a strong dependence on system size. This is as expected because these systems prefer anti-aligned spins, and so the parity of the system is critically important when it is small. The fact that these variations are not simple oscillatory ones is a result of the fact that the aspect ratio changes with $N$, which is not a consideration in the 1D case.

Note that while the computation time again scales polynomially in system size, scaling like $N^{3.3}$ on average, in this case it shows a strong dependence on the system parameters. In partic-

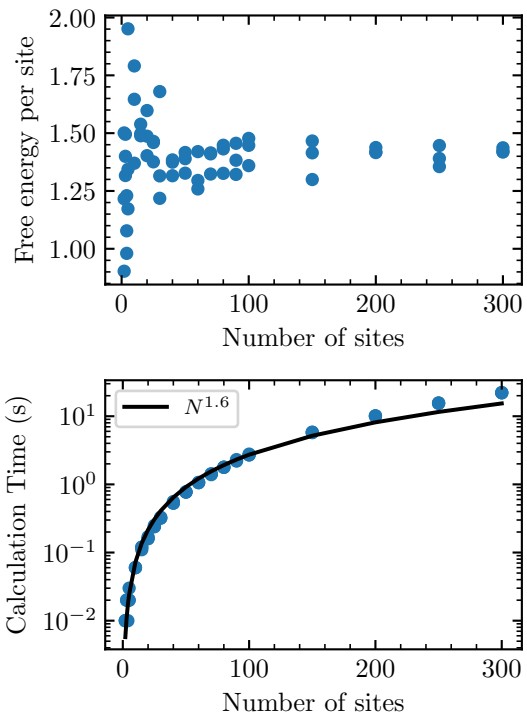

Figure 22: The free energy per site from PyTNR (top) and run time (bottom) for the 1D Ising model with disordered $h$ and $J$ is shown as a function of the number of sites. Note that three samples were drawn for each system configuration.

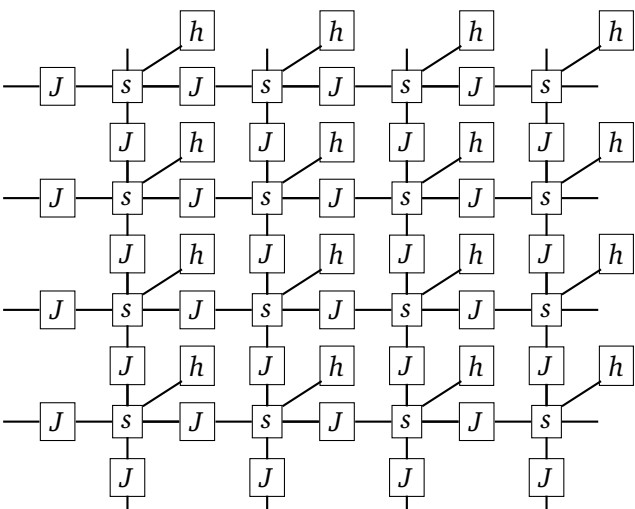

Figure 23: The 2D Ising model partition function is shown as a tensor network on a 4×4 periodic square lattice. The tensors labeled $h$ and $J$ are the same as in Figure 19, and the tensors labeled $s$ are rank 5 Kronecker delta tensors. Note that the system wraps around its edges in a toroidal fashion, so there are actually no external indices.

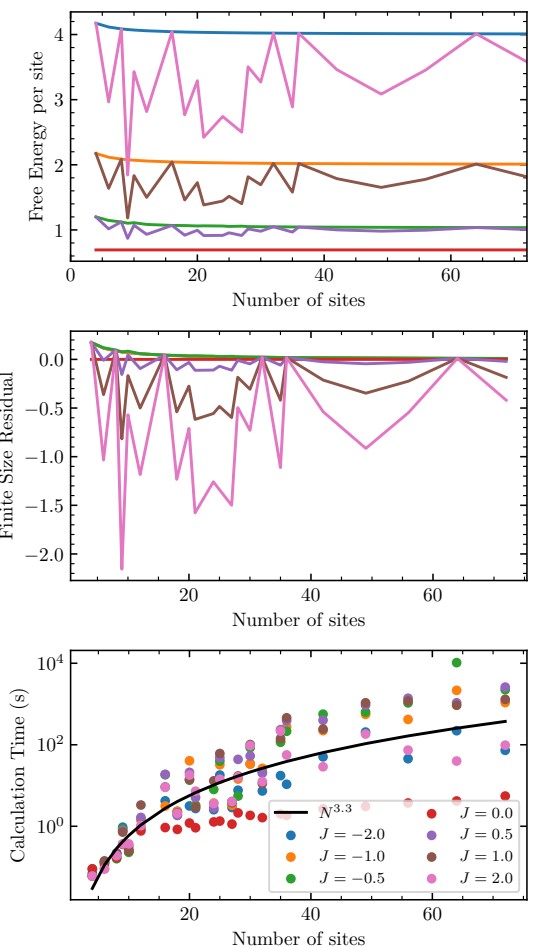

Figure 24: The free energy per site from PyTNR (top), residual versus the asymptotic value (middle) and computation time (bottom) for the 2D Ising model on a square lattice with periodic boundary conditions for $h = 0$ and several $J$ is shown as a function of the number of sites $N$. The lattice dimensions were chosen to have aspect ratios no greater than 3 and to cover a range of $N$ without large gaps.

ular, it is greatest for systems with $J$ near $J_{\text{crit}} \approx \pm 0.44$ [38], and these systems also exhibited the greatest memory requirements. This is a general feature of using tree representations in multiple dimensions: because all correlations must be transmitted through a central bond in the tree, the amount of memory required initially scales exponentially in the cross-sectional area of the system[2]. For systems larger than the correlation length this scaling is halted by the fact that different sides of the tree cease to be strongly correlated. Thus in $d$ dimensions for a system of linear size $L$ and with correlation length $\xi$, the maximum memory required during the contraction scales as

$$M \sim L^d \exp\left[\min(\xi, L)^{d-1}\right]. \tag{18}$$

This has the advantage over other methods of being asymptotically polynomial in system size, but has the disadvantage that near criticality it is effectively exponential. Near criticality it may be that MERA-type tensor representations are more performant because of their facility with long-range structure [29, 39, 40], but exploring such options is beyond the scope of this work.

---

[2]This is a perimeter in 2D, an area in 3D, and so on

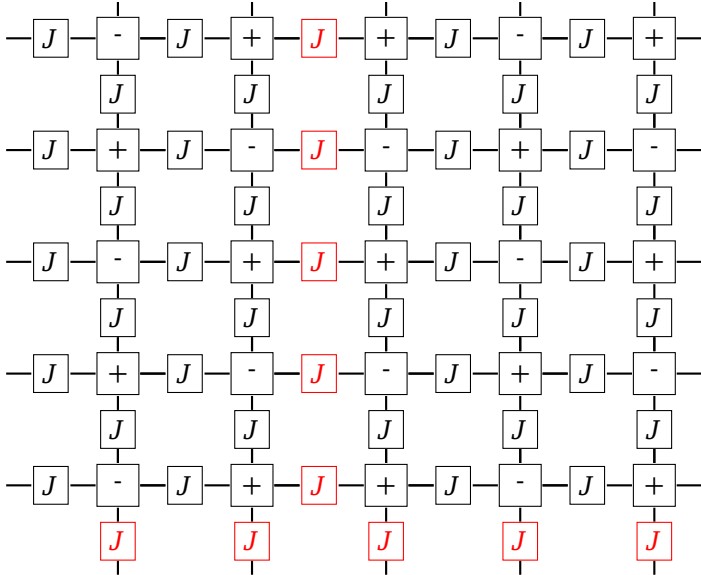

Figure 25: An example of one of the ground states of the antiferromagnetic Ising model on a 2D periodic square lattice with odd shape in both dimensions, in this case $5 \times 5$. Red boxes indicate frustrated interactions.

It is also interesting to examine the dependence of the free energy on $J$ for this model. Figure 26 shows this along with the residual versus the $N \to \infty$ limit (i.e. the finite size effects) and the computation time taken. The free energy per site is mostly symmetric in $J$, but shows a finite size asymmetry between positive and negative couplings. When $J < 0$ the system prefers all spins aligned, and so these effects are minimal. When $J > 0$ the system prefers anti-aligned spins, and so on this odd-parity lattice the residual versus the infinite system result is proportional to $J$. In particular, for a system of shape $L \times L$ with $L$ odd, $2L$ interactions must be frustrated in the ground state, as shown in Figure 25. Thus the finite size effect per site ought to be of order $2J/L$, which is what is seen.

The computation time exhibits a strong dependence on $J$. It peaks near the critical points $J \approx \pm 0.44$ because the correlations in this model are longest-ranged there, so the bond dimension must rise to accomodate the increased entanglement entropy. The model at $J = 0$ is cheap to contract because there are almost no correlations between sites.

A further informative case is the same model but with open boundary conditions, as depicted in Figure 27. Figure 28 shows the free energy PyTNR computes as a function of the number of sites and for $h = 0$ and several $J$, along with the residual versus the asymptotic result for $N \to \infty$. The same run settings were used as in the periodic case.

Unlike the periodic case, the free energy here is precisely symmetric under $J \leftrightarrow -J$. This is because the frustration which occurs when $J > 0$ in the periodic case is absent when the boundaries are open. Equivalently, in the open case the effect of negating $J$ may be undone by flipping the spins in a checkerboard pattern (i.e. letting $s \to -s$ on alternating sites). For the same reason, the finite size effects in this case are less pronounced than in the periodic case. Rather they decay in power-law fashion with system size, matching the $J < 0$ periodic cases.

Another difference worth noting is that the run time is considerably lower in the open boundary case, with a shallower scaling of $N^{2.6}$. This is because for a given near-ground state in the open system there are $\mathcal{O}(N)$ such states in the periodic system, which are generated by translations. As such tensors in the periodic system cannot be compressed as readily.

Finally, it is useful to examine the disordered case. Figure 29 shows the free energy PyTNR computes as a function of the number of sites and for $h$ and $J$ randomly and independently

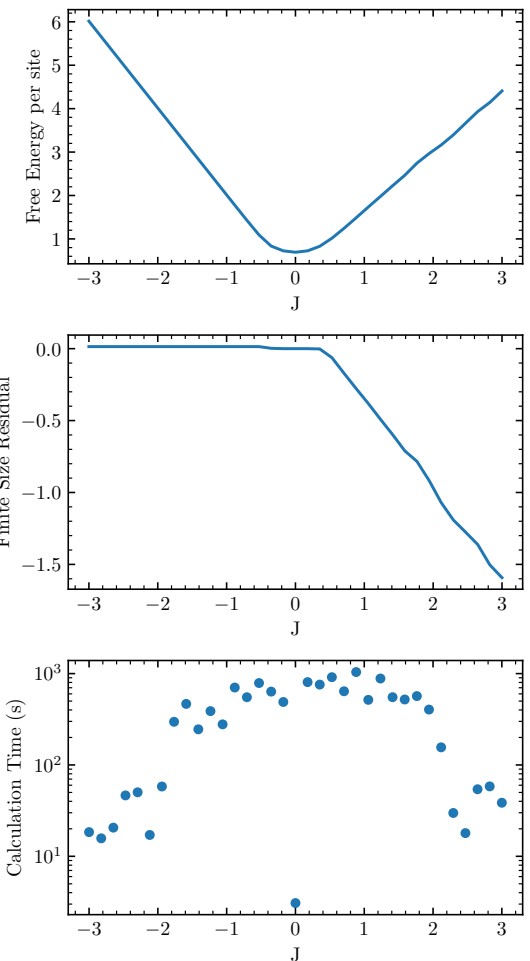

Figure 26: The free energy per site from PyTNR (top), residual versus the asymptotic value (middle) and computation time (bottom) for the 2D Ising model on a square lattice with periodic boundary conditions for $h = 0$ and as a function of $J$ for a $7 \times 7$ lattice.

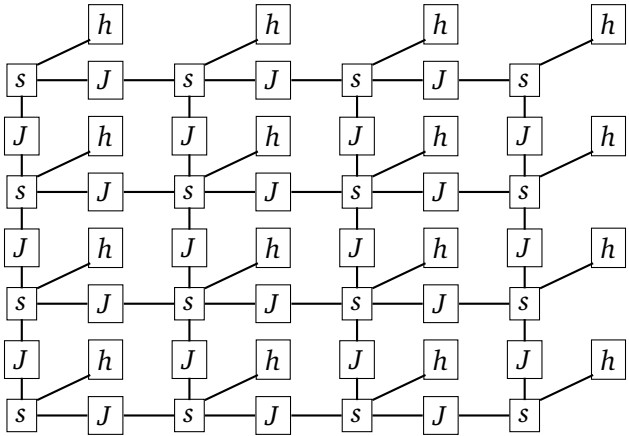

Figure 27: The 2D Ising model partition function is shown as a tensor network on a $4 \times 4$ open square lattice. The tensors labeled $h$ and $J$ are the same as in Figure 19, and the tensors labeled $s$ are rank 5 Kronecker delta tensors. Note that the system wraps around its edges in a toroidal fashion, so there are actually no external indices.

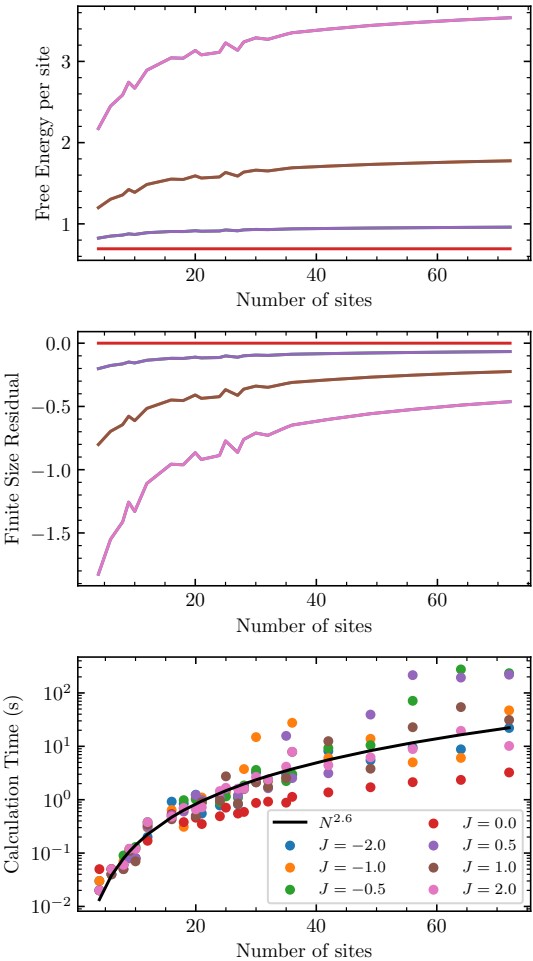

Figure 28: The free energy per site from PyTNR (top) and computation time (bottom) for the 2D Ising model on a square lattice with open boundary conditions for $h = 0$ and several $J$ is shown as a function of the number of sites $N$. The lattice dimensions were chosen to have aspect ratios no greater than 3 and to cover a range of $N$ without large gaps. Note that three samples were drawn for each system configuration.

drawn from unit normal distributions on a per-site basis. The computation time used is also shown. Open boundary conditions were used and the run settings as in the open boundary case.

Each point in this figure represents a distinct sample drawn from the distribution characterizing the model. As in the one-dimensional system the variation between neighbouring points decreases with increasing system size. This is expected because larger systems in effect average over a larger number of replicas of the system. The run time is again only weakly dependent on the sample while showing a polynomial dependence on system size.

## 6.3 Dilute Spin Glass

As a final example consider a spin glass with Hamiltonian

$$H = -\sum_i \sum_j J_{ij} s_i s_j \tag{19}$$

[41]. The model is known as dilute if only a small fraction $J_{ij}$ are non-zero [42]. There is no local structure in such systems, as any spin may be linked to any other. This makes it

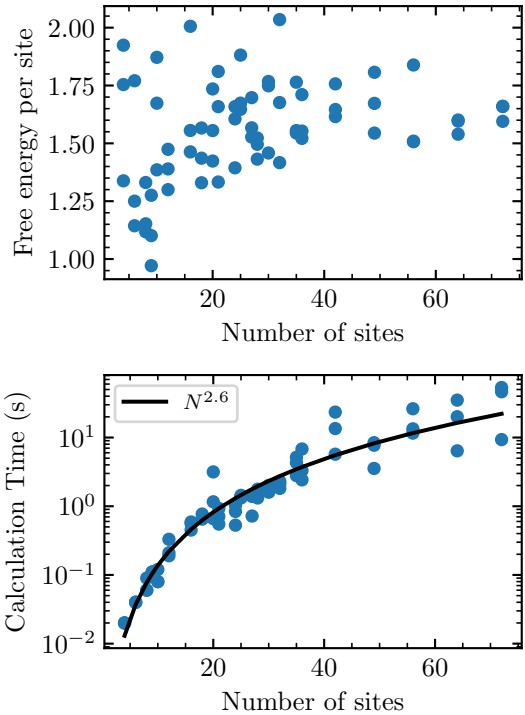

Figure 29: The free energy per site from PyTNR (top), residual versus the asymptotic value (middle) and computation time (bottom) for the 2D Ising model on a square lattice with open boundary conditions and disordered $h$ and $J$ is shown as a function of the number of sites $N$. The lattice dimensions were chosen to have aspect ratios no greater than 3 and to cover a range of $N$ without large gaps.

fundamentally dissimilar from the other networks considered here.

Figure 30 shows the free energy PyTNR computes as a function of the number of sites along with the computation time. Each sample was produced by letting a randomly chosen $\lfloor Nk \rfloor$ bonds have $J_{ij} = 1$ and the remaining bonds be zero. The bonds with $J_{ij} = 0$ were then omitted from the network.

The free energy per site shows some scatter but is remarkably constant with the number of sites and between samples. From the perspective of tensor network contraction though what is notable about this model is not the physics but the computation time, which scales strongly with the number of sites. I believe this scaling to be exponential, but I have fit both power-law and exponential relations to the timing data and see similar a similar quality of fit. If it is a power-law then it is one with an extreme exponent of roughly 10.

Whatever the precise scaling, this model proved much more challenging. It is possible that this just reflects a failure of heuristics, but given the historic challenges with these models [43] it seems possible that they do not admit a strongly compressed representation, at least not within the framework of tensor tree decompositions. This suggests that there is still much to be done in developing tensor network methods for such non-local systems.

# 7 Conclusions

I have introduced a new algorithm for contracting misaligned tensor trees and placed it in the context of a new framework for automatically contracting unstructured tensor networks.

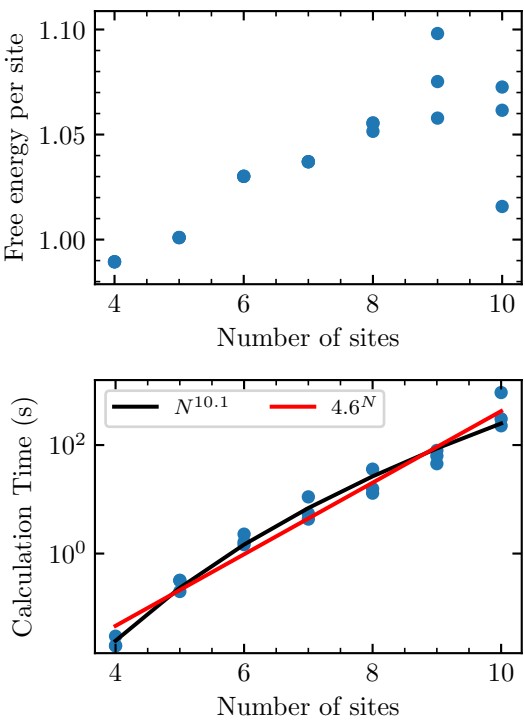

Figure 30: The free energy per site from PyTNR (top) and computation time (bottom) for the dilute Ising glass as a function of $N$. Bonds were assigned randomly to either $J_{ij} = 1$ or $J_{ij} = 0$, with a fixed number of the former equal to $\lfloor Nk \rfloor$. The samples shown used $k = 1.5$.

I have also implemented these methods in a new GPLv3 software package PyTNR. These new methods allow for rapid and controlled approximation of the contraction of finite local tensor networks off-criticality and so will hopefully enable a wider variety of investigations into both classical and quantum mechanical discrete models. Near criticality these methods fail because the tensor tree representation is inherently poor at capturing critical long-range order. This manifests in PyTNR in the form of trees with exponentially large bond dimensions near their centers, which are needed to carry area-law correlations from one side of the tree to the other. There are strong indications that this may be corrected through MERA-type representations which are designed to capture such phenomena and this provides a promising avenue for future research. Additionally, there is much to be done in developing improved contraction sequence and cycle elimination heuristics, as the ones included in this work have not been analysed in detail and different heuristics may have significantly different performance characteristics.

# Acknowledgements

I am grateful to Jesse Salomon for valuable advice on the software implementation of the methods in this work, to Milo Lin for helpful discussions regarding its physical applications, and to Jordan Cotler for thoughtful suggestions regarding spin glass models. I am also thankful for financial support from the UK Marshall Commission as well as for travel funds from both the Hertz Foundation and the Physics department at MIT. This research was funded in part by the Gordon and Betty Moore Foundation through Grant GBMF7392, by the National Science Foundation under Grant No. NSF PHY-1748958, and by the Flatiron Institute of the

Simons Foundation. This research used resources of the National Energy Research Scientific Computing Center, a DOE Office of Science User Facility supported by the Office of Science of the U.S. Department of Energy under Contract No. DE-AC02-05CH11231.

# A   Software Details

## A.1   Availability and Dependencies

The software used in this work, PyTNR (Python Tensor Network Contraction), is available under a GPLv3 license at github.com/adamjermyn/PyTNR, and consists of a Python package which implements the algorithm discussed here as well as several related methods. This package was developed for use with Python v3.5.2, NumPy v1.11.2 [44] and NetworkX-1.11, though any Python 3 distribution should suffice to run it. The plots in this work were created with Matplotlib v1.5.3 [45] and the ggplot style.

The SVD was computed using three distinct methods depending on context. For small matrices the NumPy dense SVD implementation was used. This was also used in cases where the desired rank was specified and exceeded 10% of the smallest dimension of the matrix. For large matrices with a desired rank specified the iterative SVD implementation in SciPy was used. Finally for large matrices with a desired precision rather than rank the interpolative decomposition SVD implementation in SciPy was used [46].

## A.2   Code Structure

For clarity class names in PyTNR begin with a capital letter to distinguish them from regular nouns. Thus the word 'Tensor', for instance, refers to the class in PyTNR or to an instance thereof rather than to the mathematical concept of a tensor, though the two share many similarities.

There are four fundamental objects in PyTNR: Node, Tensor, Bucket and Link. A Node is a kind of multigraph node. Each Node contains a Tensor and an ordered list of Bucket objects. This list has length equal to the rank of the Tensor in that Node.

A Bucket represents an index, and a Link is a specification of an edge in a tensor network. As such each Bucket may refer to zero or one Link objects, indicating either an external index or one which is to be contracted. Each Link object must, however, refer to precisely two Bucket objects, namely those representing the indices to be contracted.

The base Tensor class is an abstract basis class requiring Tensor objects to have shape, rank and size, along with a string representation for debugging purposes. Tensor objects must also support contraction against other Tensor objects and flattening (i.e. merging indices). There are three subclasses of the Tensor class in PyTNR. The first, the ArrayTensor class, stores and manipulates a tensor as a multidimensional NumPy array. Note that the implementation of this class monitors the norm of the array and keeps track of an exponential prefactor to avoid overflow and underflow difficulties. The second, the TreeTensor class, encodes a tensor as a tensor tree. The third, the IdentityTensor class, is a subclass of the TreeTensor class and is discussed in more detail in Appendix A.3. Finally, Tensor objects must allow setting and getting the minimal portion of their representation associated with a single index. For ArrayTensor objects this is trivial, and just amounts to manipulating the underlying array. For TreeTensor objects this amounts to addressing the leaf node corresponding to that index.

In addition to these objects, PyTNR has a Network class. A Network object contains a set of Nodes and so is a multigraph. A subclass of this is the TreeNetwork class for networks which are restricted to be acyclic.

Each TreeTensor object internally contains a TreeNetwork, and uses this to define the contraction operation as described in Section 4. By convention the Node objects inside this TreeNetwork only ever contain ArrayTensor objects, but this is a convention only. Note that contraction between TreeTensor and ArrayTensor objects is supported, and results in the latter being cast into the format of the former.

Each of these classes also contains several helper methods which are primarily used internally. There are two notable exceptions to this, namely the TreeTensor optimization and link merger methods. The former optimizes the tree to minimize memory use, detailed in Appendix A.6. This is done by performing various local operations such as index swaps between adjacent tensors and examining their impact on memory usage. The latter may be invoked when a pair of tensors share multiple indices, and in this case it flattens those indices and compresses them with a truncated singular value decomposition. In very large networks in more than one dimension this can be useful because it reduces the number of nodes in the tensor trees and thereby stops graph algorithms from being the bottleneck. The downside of doing this is that leaving the tree as is sometimes provides for a more compressed representation.

Finally there are two higher-level constructions of note, namely models and contractors. These are not classes, but rather common code patterns. Models are just methods which return tensor networks drawn from classes of interest (e.g. 1D Ising, 2D Ising, 3D Ising, etc.), and contractors are methods which control the process of contracting a network. Different contractors use different heuristics to decide on the contraction sequence and similarly use different rules for determining when to optimize tensor trees.

## A.3 Identity Tensors

A typical lattice model comes with a local state space at each site in the lattice and then defines interactions between nearby sites. A term which couples $n$ sites is most naturally expressed as a rank $n$ tensor containing the matrix elements of the interaction (i.e. the Boltzmann weights). In this language there is a high-rank Kronecker delta (identity) tensor at each site with one index per interaction term tied into that site, ensuring that every term which couples to that site sees the same state. Even for simple models the rank of this identity tensor can be large. For instance the Ising model on a $d$-dimensional square lattice has each spin interacting with $2d$ other spins, and so for $d = 2$ the identity tensor already has rank 6. Including three-spin interactions increases this dramatically, and it is easy to write down non-pathological models for which the identity tensor is too large to store directly in memory.

To circumvent this challenge, PyTNR contains a special IdentityTensor class. This is a subclass of the TreeTensor class, and directly constructs a tensor tree composed of rank 3 identity tensors. The shape of the tree is currently arbitrary, but could be specified if this proves useful.

## A.4 Cycle Elimination Heuristic

In practice a method which performs well is to define the weight of an edge connecting tensors $A$ and $B$ as

$$\text{Weight}(A, B) = \ln\left[\text{Size}(A)\text{Size}(B)\right], \tag{20}$$

where the size of a tensor is once more the number of elements it contains. In this way the weight of an edge reflects the extent to which it contributes to the overall complexity of the network, and the logarithm ensures that edges between tensors of very different sizes are not entirely weighted based on the larger of them. With this it is possible to define the minimal cycle basis, which is just the cycle basis of minimal weight (i.e. which minimizes the sum over

all cycles of the sum over all edges of the edge weight) [47]. The utility of a network is then

$$\text{Utility(network)} = \sum_{c \in \text{Minimal Cycle Basis}} \text{Length}(c), \tag{21}$$

where the length of a cycle is just the number of nodes it contains.

The minimum cycle basis is computed following the approach of [48]. That is for each edge the corresponding Horton graph is constructed. The shortest path in the Horton graph between two nodes which are adjacent in the original graph produces an element of the minimum cycle basis. The edges in this cycle are then removed from the original graph via symmetric difference, and the search proceeds until a full minimum cycle basis has been constructed.

During the cycle elimination process, the index swap which reduces the utility of the network as much as possible is chosen repeatedly until there are no cycles. I do not have a proof that this process always results in eliminating all cycles, but I have not encountered any cases in which this heuristic fails.

## A.5  Contraction Sequence Heuristics

It is difficult to identify optimal (or even acceptable) contraction sequences. This problem is tractable for small numbers of tensors when contractions are performed directly [16], but becomes extremely difficult when either the network of interest is large or the contractions are not performed directly between tensors stored as arrays.

To mediate this difficulty, several heuristics are included in PyTNR. Each of these performs well in typical use cases, but particularly near criticality the question of which one to use becomes sensitive to the problem at hand. The included heuristics, along with the relevant function names in parentheses, are:

1. Utility (utilHeuristic) - Let $U$ be the utility of the graph associated with contracting tensors $A$ and $B$, as defined in Appendix A.4, and let $M$ be the number of index pairs to be contracted between them. The contraction which maximizes $M^2/(\text{Size}(A)\text{Size}(B)\sqrt{1+U})$ is the one which is chosen at each stage. The intuition behind this is that it reflects a compromise between containing the rank explosion of the resulting tree and avoiding needlessly complex contractions.

2. Entropy (entropyHeuristic) - Let $d$ be the index dimension of one edge connecting the tensors $A$ and $B$ and let $Q$ be the number of common neighbours of $A$ and $B$ in the tensor network. The contraction which maximizes $(\text{Size}(A)\text{Size}(B)/d^2)(0.7)^Q - \text{Size}(A) - \text{Size}(B)$ is the one which is chosen. The intuition behind this is that $\text{Size}(A)\text{Size}(B)/d^2$ is the size of the tensor which will result from the contraction in a naive representation, the factor of $0.7^Q$ reduces this on the assumption that the shared neighbours are reflective of shared correlations and hence a propensity for compression, and the final two terms favor contracting bigger objects, all else being equal.

3. Merge (mergeHeuristic) - This heuristic performs the first contraction, if any, that it can find which involves a tensor of rank at most 2, as there is no danger of rank explosion with such tensors. It does this until there are no more such tensors to be contracted, at which point it contracts the pair of tensors with the most common neighbours. This reflects the intuition that tensor pairs with more common neighbours have more redundant correlations between them and hence the final result will be more readily compressed.

4. Small Loop (smallLoopHeuristic) - This heuristic performs the first contraction, if any, that it can find which involves a tensor of rank at most 2, as there is no danger of rank

explosion with such tensors. It does this until there are no more such tensors to be contracted, at which point it contracts the pair of tensors which minimizes Rank($A$) + Rank($B$) − $Q$ + $W$, where $Q$ is as defined previously and $W$ is the length of the largest cycle between the two tensors. This reflects the intuition that tensor pairs with more common neighbours have more redundant correlations between them whilst avoiding excessively complex contractions.

5. Loop (loopHeuristic) - At each stage this heuristic performs the contraction which maximizes the greatest distance within either tensor tree between any pair of indices being contracted. Tensors which are not represented by tensor trees are assigned a distance of 100, which means that they will be contracted first under most circumstances. This is meant to prevent cycles from becoming large in the first place.

6. One Loop (oneLoopHeuristic) - This provides an alternate implementation of the Loop heuristic.

The Merge, Loop and Entropy heuristics are the most well-tested and are recommended unless there is a context-specific reason to prefer one of the others.

Note that the term "replica" in front of a heuristic name indicates that that heuristic is used with the replicaContractor method. This method maintains a user-specified number of replicas of the tensor network. In cases where the heuristic views multiple different contraction choices as equally good, this contractor will select one of them at random and apply it to the replica with the lowest memory footprint. In this way, choices which result in large increases in memory usage need not be pursued further unless such increases appear to be inevitable, either because there is only one choice available with the given heuristic or because a large fraction of the available choices also yield increasing memory usage.

## A.6 Tree Optimization

Tree optimization is done in three stages. First, all rank 2 tensors in the tree are contracted against rank 3 tensors, if possible. This just reduces the number of tensors which need to be considered and stored. Secondly, all tensors which are doubly linked to one another are contracted. This is just an extension of the first stage because such tensors have effective rank 2.

At the end of the second stage, every internal link in the tree is of the form

$$A_{ijk}B_{klm}. \tag{22}$$

The same object may also be written as

$$C_{ilk}D_{kjm} \tag{23}$$

or

$$E_{imk}F_{kjl}. \tag{24}$$

These possibilities are depicted in Figure 31. These three representations generally require different internal index dimensions because they co-locate different pairs of indices. It is more efficient to co-locate highly correlated indices, and so one of these is generally preferable. The algorithms which generate tree structures during the contraction process are not guaranteed to make the optimal choice for each pair of neighbouring tensors, and so this must be enforced afterwards. This is what the third optimization stage does.

To do this, PyTNR marks each Link in the TreeTensor as 'not done'. It then picks one which is marked as such and picks the optimal configuration. If this is the configuration it was

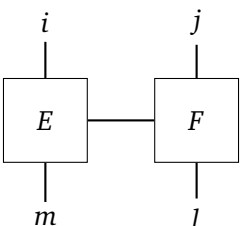

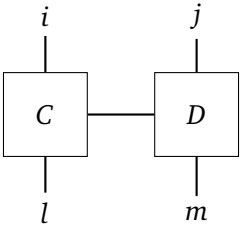

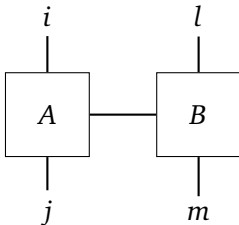

Figure 31: Three representations of the same tensor network are shown. In each there is a single contraction over a single pair of indices between the two tensors, as well as two external indices on each. These are all of the unique factorizations of a rank 4 tensor into a network of two rank 2 tensors.

already in it just marks that Link as 'done'. Otherwise it marks that Link as 'done' and marks all other Link objects connected to either of the newly created Node objects as 'not done'. This process proceeds until all Link objects are marked as 'done', at which point the optimization is complete.

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
