# Peer review of "Automatic Contraction of Unstructured Tensor Networks"

_SciPost Physics, doi:SciPost Phys. 8, 005 (2020)_

## Round 3 · Referee Report · Anonymous · 2019-9-24

Strengths

1 A possible algorithm for contracting a cyclic tensor network without rank increase is suggested.

2 A part of the problem presentation is well written and the suggested algorithm is precisely explained.

Weaknesses

1 The efficiency of the proposed algorithm is not sufficiently discussed.

Report

In a tensor network, the rank explosion after successive contraction operations is often serious problem for a practical computation. In this manuscript, an algorithm for contracting unstructured (or cyclic) tensor networks, which may be originated from misaligned-tree structure, without the rank increasing is proposed. The rank explosion problem as a motivation of this study is well explained, and the proposed contraction (or unraveling) procedure using the singular value decomposition is carefully described.

The idea would to be highly suggestive and interesting, although there is still plenty room for improvement about the contraction sequence. However, the efficiency of suggested algorithm seems not to be sufficiently demonstrated because most of the descriptions on the results of numerical experiments are devoted to discussions on the finite-size effects. I agree that the finite-size effects are physically interesting; nevertheless, in this paper the efficiency of using the suggested algorithm should be more important and clearly written, namely, benefit and limit of the suggested algorithm in the performed calculations. Before further consideration of this manuscript, I would like the author to revise the numerical experiments part and also address the following questions and suggestions.

I have some questions and suggestions about the numerical experiments:
- What is the real meaning of residual for the 1D Ising model? Does the middle panel of Figure 20 simply show the machine floating point precision? Why is the residual shown in Figure 21 ($J=0$) much larger than that shown in Figure 20 ($h=0$)? It is because the case of $J=0$ looks much simpler than the case of $h=0$.
- In the periodic 2D Ising model, the finite-size effects strongly depend on either the system is (even)$\times$(even) or (even)$\times$(odd) or (odd)$\times$(odd). For the latter, the difference between the cases of $J>0$ and $J<0$ is larger. This should be mentioned around the description on Figure 24.
- About the secondary peak at $J \approx 1.5$ in the bottom of Figure 26, the author wrote "it is most likely a result of a poor choice of contraction sequence’’. How did the author arrive at that conclusion? If this is true, is there a way to solve this problem?
- On the description starting with "Unlike the periodic case, the free energy here is...’’, the reason why the results between $J>0$ and $J<0$ match is simply that a frustration as discussed in Figure 25 never occur even for $J>0$ in open cluster. Such an explanation may be helpful for reader especially from the field of condensed matter physics.
- In Conclusions, it is written as "Near criticality these methods fail because the tensor tree representation is inherently poor at capturing critical long-range order’’. Can any signatures about this statement be found in the performed numerical experiments?

Minor questions/suggestions:

(i) Isn’t Eq.(13) correctly $F=-\ln[(2\sinh(J))^N+(2\cosh(J))^N]$?
(ii) It would be helpful for reader to write the definitions/meanings of $m$ and $n$ in Eqs. (3)-(7) before and after eliminating singular values below the threshold.
(iii) In the top and middle of Figure 27 only three or four lines are visible. Are the lines for $\pm J$ perfectly overlapped? And, is the caption correct (is it disordered?)?
(iv) I think that the spin $s_i$ is generally defined as not tensor but coordinates of tensor in the Ising models. While in this paper, it is regarded as the Kronecker delta tensors. Is it possible to explain it in a little more detail or cite a reference on it?
(v) Is the caption of Figure 29 correct? Isn’t it for the disordered case?

Requested changes

1 Revision of the numerical experiments part to further discuss the efficiency of proposed algorithm.

2 Further revisions with addressing my questions/suggestions.

---

## Round 3 · Referee Report · Anonymous · 2019-10-1

Strengths

1 - very clearly written
2 - introduces an interesting algorithm for evaluating partition functions
3 - illustrative examples, showing the strengths and weaknesses of the algorithm

Weaknesses

Not many weaknesses, maybe stronger relation to existing tensor network techniques possible.

Report

The manuscript discusses a way to evaluate partition functions of discrete (lattice) systems by contracting tensor networks. To do so, a new algorithm is introduced, which relies on restructuring the index map of tensors such that matrices are obtained, which then can be modified/truncated using singular value decomposition. This is very reminiscent of tensor network techniques (like matrix product states, see, e.g., arXiv:1008.3477), and it might be useful for non-expert readers to make a stronger relation to this literature.

I enjoyed reading the paper, as it is extremely well written, and I think the observation made by the author will be helpful in further developing tensor network techniques, so that after considering optional changes the manuscript can be published.

Requested changes

1- stronger relate the procedure described on page 10 to tensor network methods, like matrix producht states, where reshaping tensors and applying the SVD (as is done here) is at the heart of the methods.
2- on page 15, below Eq. (14) the statement is made that "...this trend is a polynomial in system size...", without further justification. It would be helpful for the reader to find a short reasoning for why the author believes this is a polynomial dependence.

---

## Round 4 · Referee Report · Anonymous (Referee 1) · 2019-11-23

Strengths

1 A possible algorithm for contracting a cyclic tensor network without rank increase is suggested.
2 A part of the problem presentation is well written and the suggested algorithm is precisely explained.
3 The computational requirements for the proposed algorithm are clearly evaluated. The software (PyTNR) used in this work is available.

Weaknesses

no obvious weaknesses (dare I say it, the performance for quantum systems is unknown but I hope it would be studied in future).

Report

I am really satisfied with the revised manuscript. The main reasons are the followings:

-The presentation of computational results, especially the figures, has been much more sophisticated and the efficiency of proposed algorithm is now clearly understandable.

-The additional description on the replica contraction sequence is very useful and it makes the computational performance more stable with avoiding a bad choice of contraction sequence.

-The manuscript has been sufficiently improved by incorporating both of the referee’s suggestions.

Therefore, I recommend the publication of this paper in the present form.

Requested changes

Nothing

---

## Round 4 · Author Response

Dear Editor and Referees,

I am grateful for the very helpful uggestions by the referees and have incorporated them in a revised manuscript.

To make the computational requirements of PyTNR clearer I have extended the numerical experiments to larger system sizes and compared the timing results to different scaling laws. Based on this I estimate that for 2D systems contraction scales roughly like the number of sites cubed. I have incorporated this into the figures and discussion. I have also incorporated improvements to some of the heuristics into the appendices.

A point-by-point response is included below.

Again thank you for your comments, and I hope you find this revised manuscript suitable for publication.

Adam S. Jermyn

---

## Round 4 · List of Changes

Referee 1:

1- stronger relate the procedure described on page 10 to tensor network methods, like matrix producht states, where reshaping tensors and applying the SVD (as is done here) is at the heart of the methods.

This is a good point, and I have added a paragraph to that section discussing the relation between the cycle elimination algorithm and MPS optimization via SVD.

2- on page 15, below Eq. (14) the statement is made that "...this trend is a polynomial in system size...", without further justification. It would be helpful for the reader to find a short reasoning for why the author believes this is a polynomial dependence.

This claim is based on the scaling shown in the third panel of Figure 20 which appears close to a logarithmic dependence in the log-linear scaled plot. That indicates a power-law scaling. To make this clearer I have fit the runtime data in each case to a power-law, overlayed that fit, and included the exponent of the power-law in the figure legend. In all but one case there is a good fit to a power-law with exponent of at most four. The exception is the example in Figure 30 of a spin glass, which has an exponent of 10. That large exponent suggests that in that case the runtime is actually exponential. To test this possibility I have also fit an exponential to the timing data in that case and found it to be comparable consistent to the power-law fit.

Note also that power-law behavior is expected for transfer matrix methods, and for 1D Ising models my methods are very similar to these. I have added a discussion of this point to the text around page 15.

Referee 2:

  • What is the real meaning of residual for the 1D Ising model? Does the middle panel of Figure 20 simply show the machine floating point precision? Why is the residual shown in Figure 21 (J=0) much larger than that shown in Figure 20 (h=0)? Is it because the case of J=0 looks much simpler than the case of h=0?

Yes, the middle panel of Figure 20 just shows machine floating precision. This is because the contraction heuristic determines The greater residuals in Figure 21 are because, unlike the case where h=0, when J=0 the heuristics end up using the singular value decomposition. In an updated version of PyTNR this does not happen, and I have redone the calculations with the newer version to avoid confusion.

  • In the periodic 2D Ising model, the finite-size effects strongly depend on either the system is (even)×(even) or (even)×(odd) or (odd)×(odd). For the latter, the difference between the cases of J>0 and J<0 is larger. This should be mentioned around the description on Figure 24.

I have added some discussion of this point to the text around Figure 24.

  • About the secondary peak at J~1.5 in the bottom of Figure 26, the author wrote "it is most likely a result of a poor choice of contraction sequence’’. How did the author arrive at that conclusion? If this is true, is there a way to solve this problem?

I have seen poor choices arise in various experiments with PyTNR, and they can be identified by trying different contraction heuristics and seeing that some exhibit peaks in runtime and others do not. To address this point I have created a ``replica contraction sequence'' in a new version of PyTNR which attempts several different contraction sequences, not just the one the chosen heuristic prefers the most. The manager works on the sequence with the smallest memory footprint at any given time, which avoids many cases where a bad choice forces the contraction process into a sequence of expensive steps. I have detailed this new software layer in the appendix and re-run the numerical experiments, using it in all 2D contractions. As the updated figures show, there is no longer a spike in the runtime cost near J=1.5.

  • On the description starting with "Unlike the periodic case, the free energy here is...’’, the reason why the results between J>0 and J<0 match is simply that a frustration as discussed in Figure 25 never occur even for J>0 in open cluster. Such an explanation may be helpful for reader especially from the field of condensed matter physics.

This is a good point and I have added this explanation to the text.

  • In Conclusions, it is written as "Near criticality these methods fail because the tensor tree representation is inherently poor at capturing critical long-range order’’. Can any signatures about this statement be found in the performed numerical experiments?

I have added some text to the conclusions explaining that this becomes apparent through increasing (exponential) contraction times, with very large bond dimensions in the central bonds of the trees.

  • (i) Isn’t Eq.(13) correctly F = -ln[(2 sinh(J))^N + (2 cosh(J))^N]?

Indeed the referee is correct and I have fixed this in the text.

  • (ii) It would be helpful for reader to write the definitions/meanings of m and n in Eqs. (3)-(7) before and after eliminating singular values below the threshold.

I have added this.

  • (iii) In the top and middle of Figure 27 only three or four lines are visible. Are the lines for +-J perfeclty overlapped? And is the caption correct (is it disordered)?

The caption was incorrect and has been fixed. The lines for +- J indeed overlap perfectly as mentioned in the text.

  • (iv) I think that the spin s_i is generally defined as not tensor but coordinates of tensor in the Ising models. While in this paper, it is regarded as the Kronecker delta tensors. Is it possible to explain it in a little more detail or cite a reference on it?

I have added some more discussion of this to the text in section 6.1 where I first introduce this notation, as well as to the caption of Figure 19.

  • (v) Is the caption of Figure 29 correct? Isn’t it for the disordered case?

It is for the disordered case and I have corrected this.

---

## Editorial Decision

published